# Sweep Sampling Comparison of Terrestrial Insect Communities Associated with Herbaceous Stratum in the Riparian Zone of the Miho River, Korea

**DOI:** 10.3390/insects13060497

**Published:** 2022-05-25

**Authors:** Jeong Ho Hwang, Mean-Young Yim, Sung-Yeol Kim, Seong Jin Ji, Wang-Hee Lee

**Affiliations:** 1Natural History Division, National Science Museum of Korea, Daejeon 34143, Korea; myyim90@nsmk.kr (M.-Y.Y.); jsj76@korea.kr (S.J.J.); 2Department of Smart Agriculture Systems, Chungnam National University, Daejeon 34134, Korea; wanghee@cnu.ac.kr; 3Research Division, Enfield Co., Daejeon 34134, Korea; sdzzangksr@naver.com

**Keywords:** insect sampling, riparian zones, herbaceous stratum

## Abstract

**Simple Summary:**

Insect pests and their natural enemies can harbor in riparian zones. To determine the impact of insect communities on agriculture and ecology, we must quantitatively assess insect populations in riparian areas. To identify the appropriate methodology for effective insect sampling in riparian areas, we assessed sweep sampling within three plant communities using different numbers of subsampling units (50 sweeps carried out twice, or 10 sweeps over 10 times) over two years. The results reveal that effective insect sampling varies between different plant communities and insect orders. The similarities between terrestrial insect communities in the same plant community were relatively high, even in different years. The optimum sampling size to obtain approximately 80% of the total species was estimated for each survey site. Our results lay the foundations for providing techniques to assess insect populations within riparian areas to predict and prevent herbivorous insect pest invasions in the future.

**Abstract:**

To investigate insect and plant community relationships in riparian zones, terrestrial insect communities were compared in plant communities in the riparian zone of the Miho River, Korea. The sweep netting method was used to sample insects in 50 m transects in three herbaceous plant communities. In 2020, each plant community—*Chenopodium album*, *Beckmannia syzigachne*, and *Artemisia indica*—was swept 100 times (50 sweeps × 2). In 2021, two communities had an additional 100 sweeps collected using 10 subsamples of 10 sweeps (excluding *C. album* communities). The surveyed dominant species or subdominant species of the insect community in each site preyed on the dominant plant species at the site. The Bray–Curtis similarity was significantly higher than the Sørensen similarity when comparing datasets across different years for the same plant species community. The predicted optimum sampling size to obtain approximately 80% of the total species estimated to be at each survey site, for effective quantitative collection of terrestrial insect herbivores in each plant community, was examined. Fifty sweeps were required for the *A. indica* community and 100 sweeps were required for the *B. syzigachne* community. The results of this study provide important data for riparian biodiversity conservation and future pest monitoring.

## 1. Introduction

Riparian zones are the areas of transition and interaction between terrestrial and aquatic environments [1]. These areas, in their natural state, are considered the most dynamic and diverse habitats in a terrestrial environment [2]. Studies have been conducted with herbivorous insect communities and specific taxa that have a high ecological role, such as leaf-litter ants, to evaluate biodiversity in riparian zones [3,4,5]. Ecologically important riparian zones are mainly conserved and managed with a policy of designing and maintaining a buffer zone [6].

Plants provide habitat for most insects; however, many insect species have a strong specificity for the environment and host plants [7]. Insects and plants have been studied in combination, for a variety of purposes, in community ecologies. Many studies have focused on understanding the characteristics and correlations between plant and insect diversities [8,9,10]. In the context of agricultural pest management, insect communities in plant communities near agricultural fields have been investigated [11,12]. In Korea, insects and the host plants of major pests have been investigated in fallow paddy vegetation to identify the ecologies of pests, as well as those of potential herbivore insects for weed control [13,14,15]. However, studies combining insect herbivore host plants with the flora of a surveyed area to investigate insect and the plant community relationships are limited, especially in riparian zones [3].

As it is impractical to collect all the insect species at a survey site, effective use of quantitative sampling methods can help in comparing regional biodiversity by providing a snapshot sample of the species residing in plants [16,17]. Although sweep sampling has been evaluated previously, its use in comparing insects in different plant communities is limited [16,18,19]. In this study, quantitative sweep sampling results for each plant community were used to estimate the optimum sampling size, including the sampling rate for the estimated number of insect species.

Sweeping has been compared with several insect sampling methods and used to evaluate insect communities in agricultural and natural ecology [20,21]. In natural ecology, it is used to determine seasonal and regional biodiversity, as well as energy flow evaluation [22,23,24]. In agricultural ecology, sweeping has been used to determine pest and natural enemy population dynamics, including emigration, and pesticide efficiency [25,26,27]. Quantitative sweep sampling for pests has also been used to determine the relevance of a population to other taxa, such as avian species [28].

This study focuses on the characteristics, correlations, and similarities of insect communities in riparian plant communities in Korea. We assumed that the similarities of insect communities in the same plant community would be high, even in different years. We aimed to provide the information needed to conserve diversity by analyzing the differences in insect communities according to plant communities in the riparian zone of the Miho River and to create data to help optimize insect sampling.

## 2. Materials and Methods

### 2.1. Survey Site Characteristics

The Miho River is known for its ecologically diverse environment, providing habitats for various plants and fish; very limited research has been conducted on insect populations [29,30,31,32]. The Miho River tributary joins the Geum River across the western Chungcheongbuk-do, including Jincheon-gun and Cheongju-si. The Jincheon and Miho Plains and extensive agricultural land are distributed across the landscape near the river [33]. In addition to paddies and farms, there are several parks and parking lots near the river that result in anthropogenic impacts on the river. The river also experiences natural disturbances, such as frequent flooding during the rainy season.

Meteorological data were obtained from the Cheongju Observatory (No. 131, 36.63924 N, 127.44066 E, 58.7 m above sea level) near the survey sites. In the spring (March–May) of 2020, the average temperature was 13.2 °C, with average minimum and maximum temperatures of 7.9 °C and 19.2 °C, respectively, and average precipitation of 103.9 mm. In the spring of 2021, the average temperature was 14.1 °C, with average minimum and maximum temperatures of 9 °C and 19.7 °C, respectively, and average precipitation of 259.4 mm [34].

Survey sites were selected in the pre-flooding season for each of the dominant plant communities to identify correlations between the insect and plant communities of the Miho River. The following plant communities were selected: the *Chenopodium album*, *Beckmannia syzigachne*, and *Artemisia indica* communities. The survey locations were restricted to areas where the selected plant communities encompassed at least 50 m of the riparian zone of the Miho River. The terrestrial insect community and the surrounding vegetation were investigated in all the survey sites. The survey sites of the *A. indica*, *B. syzigachne*, and *C. album* communities were present in the upstream, midstream, and downstream areas of the Miho River, respectively.

The *C. album* community survey site (36.598806 N, 127.342013 E) was approximately 146 m from the open water zone; the river width at the site was approximately 120 m and the river terrace was widely formed (approximately 417 m). *Humulus japonicus* were densely distributed along the riverbank. Moreover, a village was present across the river, with rice paddies and other crop fields on the western side of the river. The soil type of the riparian area was mostly sandy; therefore, it was a dry environment with good drainage. Flooding occurs frequently in this low-lying area during the rainy season.

The *B. syzigachne* community survey site (36.734940 N, 127.483070 E) was located approximately 27 m from the river channel; the river width at the site was approximately 110 m and the width of the river terrace was approximately 75 m. This low-lying survey site also floods frequently due to its proximity to the river. The surrounding vegetation included *C. album* and *Salix koreensis* communities. There were rice paddies and farms along the road across from the river.

The *A. indica* community survey site (36.936856 N, 127.461964 E) was approximately 15 m from the river channel; the river width at the site was approximately 70 m. The riverbank, where the *A. indica* community was formed, was narrow (8 m) and steep. The site was adjacent to the road next to the village to and crop fields, and experienced light flooding due to the high embankment. Moreover, due to the site’s close proximity to the road, weeding was periodically conducted in the summer season (Figure 1).

### 2.2. Survey Methods

To investigate the plant community in the riparian zone of the Miho River, a field survey was conducted in the spring (i.e., in May in 2020 and 2021), prior to the rainy season. In the 2021 field survey, the *C. album* community had not yet developed due to flooding in 2020; therefore, surveys were conducted only for the *B. syzigachne* and *A. indica* communities on 18 May. In 2020, surveys were conducted on 13–14 May between 10:00 and 14:00. Sweep sampling (100 sweeps; 50 m) was conducted for quantitative analysis of the terrestrial insect communities in each plant community. In 2020, sampling was divided into two subsampling units of 50 sweeps each (25 m). In 2021, samples were collected from 10 subsampling units of 10 sweeps each (5 m). The insect sweep net had a total length, depth, and diameter of 1200, 700, and 300 mm, respectively.

The flora investigation involved 50 m set transects at each site where the insect surveys were undertaken. Plants 2 m to the left and right of each survey transect were recorded. The dominant plants around the survey transect were also recorded. The Braun-Blanquet method was followed to conduct the vegetation survey. For the plant community nomenclature, we followed the National Ecosystem Survey Guideline of Korea, which names the community based on the plant species dominating over 70% of the top-layer species [35,36].

### 2.3. Identification of Plant and Insect Species

Plant species were identified from previous literature [37,38,39]. The plant classification system was based on the National Standard Plant List [40] and the 2020 National List of Species [41]. The insect samples were identified after sorting, using a stereomicroscope (Olympus SZ1145; Olympus, Tokyo, Japan). Various illustrations and articles were referred to for the identification of species, and experts were consulted for any unidentified species [42,43,44,45,46,47,48,49]. The insect classification was based on the Korean Insect List (1994) and the 2020 National List of Species [41,50]. The insects collected were stored in 94% ethanol or retained as dried specimens in the Natural History Laboratory of the National Science Museum. The known host plants of the herbivorous insects sampled and the plant fauna at each site were compared and matched, based on previous literature [13,42,43,44,45,46,49,51,52,53,54,55,56,57,58,59,60,61,62,63,64,65].

### 2.4. Community Structure Analysis

To determine the diversity index, we used the modified formula by Lloyd and Ghelardi from the Shannon–Wiener function index derived from Margalef’s information theory [66]. Dominance, richness, and evenness indices were calculated using McNaughton’s dominance index, Margalef’s index, and Pielou’s formula, respectively [67,68,69]. The similarity indices were calculated using the SPADE program with the Sørensen index, an incidence-based similarity index. The Bray–Curtis index provided an abundance-based similarity index that is effective regardless of limited sampling errors [70,71,72,73,74]. We used the clustering method (unweighted pair group method with an arithmetic mean), using IBM SPSS Statistics (IBM Statistics, Chicago, IL, USA). We produced sample size-based (abundance or sampling units) rarefaction and extrapolation sampling curves to calculate the species richness of each community and each year. The statistical analyses were conducted using iNEXT in R software version 3.6.0 (R Foundation for Statistical Computing, Vienna, Austria). The curves were made using 95% confidence intervals and the reference sample size was doubled [75,76].

## 3. Results

### 3.1. Analysis by Site and Year

A total of 4491 terrestrial insects belonging to 5 orders, 42 families, and 109 species, were collected in the 2020–2021 surveys. In 2020, 417 insects (from 2 orders, 10 families, and 18 species), 520 insects (from 4 orders, 15 families, and 29 species), and 1343 insects (from 4 orders, 14 families, and 24 species) were collected from the *C. album, B. syzigachne,* and *A. indica* communities, respectively. In 2021, 284 insects (from 4 orders, 18 families, and 30 species) and 1927 insects (from 4 orders, 29 families, and 59 species) were collected from the *B. syzigachne* and *A. indica* communities, respectively.

The site-specific terrestrial insect community analysis revealed that, in 2021, the *A. indica* community had the highest diversity index (H’), richness index (RI), and evenness index (EI), while the *B. syzigachne* community had the highest dominance index (DI). In 2020, *Psylliodes attenuata* was the dominant species in both the *C. album* and *B. syzigachne* communities. In 2021, *Capsus cinctus* was the dominant species in the *B. syzigachne* community, while in 2020 and 2021, *Austroasca vittata* was the dominant species in the *A. indica* community. Matching the known host plants of the collected terrestrial insect herbivores with the plant flora of each site and each year revealed a matching rate of at least 66.7%. The matching rates for *C. album, B. syzigachne,* and *A. indica* communities in 2020 were 100, 82.6, and 87.5, respectively; for both *B. syzigachne* and *A. indica* communities in 2021, the matching rate was 66.7 (Table 1 and Appendix A).

### 3.2. Comparison of the Number of Species and Individuals by Order in Each Site and Season

The numbers of insects belonging to the orders Hemiptera and Coleoptera were the highest at all sites. However, large numbers of individuals from Diptera and Hymenoptera were also found in *A. indica* communities (Table 2).

### 3.3. Similarity Analysis

In 2020, similarity results were analyzed and compared for 50 sweeps for the initial 25 m and 50 sweeps for the final 25 m, for a total of 100 sweeps over 50 m. The Sørensen incidence revealed a similarity of 0.62 or higher in the same plant community (Figure 2, Appendix A).

The Bray–Curtis similarity can be divided into either ≥0.1 or <0.1. Basically, insect community similarities in the same plant community were high. In addition, the similarity was relatively high in the *C. album* and *B. syzigachne* communities. The other cases had a Bray–Curtis similarity below 0.1 (Figure 3, Appendix A).

We compared the similarity with the years and plant communities using sweep analysis data—2 units of 50 sweeps in 2020 and 10 units of 10 sweeps in 2021—to compare the 100 sweep units. The *B. syzigachne* community had the highest Sørensen incidence similarity index, with a score of 0.34, while the *A. indica* community had the highest Bray–Curtis similarity, with a score of 0.62 (Figure 4, Appendix A). 

### 3.4. Species Diversity Abundance Curves of the Insects According to Plant Communities in 2020 and 2021

Abundance-based species diversity curves for the number of insect species in each plant community were estimated using the combined results from 2020 and 2021. The annual and plant community curves were also estimated. For the same number of individuals, species diversity was the highest in the *B. syzigachne* community and in the *A. indica* community (Figure 5).

### 3.5. Sampling Method Impact on Species Diversity and Insect Sample Coverage Curves by Plant Communities in 2021

In 2021, sampling was conducted in *B. syzigachne* and *A. indica* communities (each with 10 sweeps for 10 subsampling units). The sampling units-based species diversity and sample coverage curves for the insects were estimated. The *A. indica* community had a higher number of species per unit sample than the *B. syzigachne* community. The ratio of the sampled species to the estimated species number revealed that only 5 subsampling units (50 sweeps) were required for the *A. indica* community to produce 81.2% sample coverage. In contrast, for the *B. syzigachne* community, 10 subsampling units (100 sweeps) were required to produce a 78.8% sample coverage (Figure 6).

### 3.6. The Coefficient of Variation of the Insect Samples in 2021

In 2021, the coefficient of variation was analyzed using the results of the 10 sampling units from 10 sweeps. The difference between coefficients of variation for the two communities was greatest for the Hymenopteran species. The Hemipteran species in the *A. indica* community had the lowest coefficient of variation (Table 3).

## 4. Discussion

Insect sampling in the plant communities of Miho River revealed that the number of species more than doubled (to 59 species) in the *A. indica* community in 2021, compared to 2020 (24 species). Meteorological data for 2020 and 2021 revealed that the average temperature and precipitation in spring (March to May) was 13.2 °C and 103.9 mm and 14.1 °C and 259.4 mm, respectively. The higher temperature and precipitation in 2021 may have contributed to the increase in the number of insect species in the *A. indica* community in 2021 [77]. The *B. syzigachne* community was impacted by flooding in 2020, which may have had a negative effect on the species diversity, reducing the positive effect of favorable meteorological conditions in 2021 and resulting in a relatively constant number of species [78].

A previous study conducted in May 2019 on herbivorous insects in the Gap River revealed 9, 10, and 10 species in the upper, middle, and lower stream sites, respectively [3]. Our study in the Miho River revealed 13, 23, and 16 species in the *C. album*, *B. syzigachne*, and *A. indica* communities in 2020 and 2021, respectively, and 21 and 34 species in the *B. syzigachne* and *A. indica* communities in 2020 and 2021, respectively. Therefore, species diversity was higher in the Miho River than in the Gap River [3]. However, unlike the survey sites on the Miho River, which were primarily in rural areas, the survey sites of the Gap River encompassed the city center, thus experiencing greater human interference that may have affected the results. In addition, urbanization index analysis, which evaluates the degree of human interference with the naturalized plants in the survey area, revealed that the Gap River had an urbanization index of 14.8, which was much higher than that of the Miho River (5.2) [79,80]. The chemical composition of the vegetation at the sites may also affect the insect community [81].

Among the three plant communities surveyed, the *C. album* community had the highest number of plant taxa (26 taxa), but it had fewer insect species than the *A. indica* or *B. syzigachne* communities. It is likely that the intrinsic characteristics of the vegetation, including plant chemicals and other environmental factors rather than the number of plant taxa, influenced the number of insect species [81].

*Psylliodes attenuata* was the dominant species in the *C. album* community, due to the high density of *Humulus japonicus* (*P. attenuata* host) in this community [49]. *Orthotylus flavosparsus*, a known pest of sugar beet and quinoa, was the subdominant species in the *C. album* community; it feeds on *Chenopodiaceae*, including *C. album* [42,82].

*P. attenuata* was the dominant species in the *B. syzigachne* community in 2020 (similar to its dominance in the *C. album* community), but it did not reappear in 2021, possibly due to the decrease in the density of *H. japonicus* in 2021 because of flooding. Moreover, three weevil species, *Psilarthroides czerskyi*, *Cardipennis shaowuensis*, and *Cardipennis sulcithorax*, that feed on *H. japonicus* were collected in 2020 but were not recorded in 2021 [45]. In 2020, the subdominant species in the *B. syzigachne* community was *Capsus cinctus*, which became the dominant species in this community in 2021. The genus *Calamagrostis* and the species *Eleusine indica* from the *Poaceae* family are the recorded host plants of this species. We confirmed that the species also consumed *B. syzigachne,* which is also from the *Poaceae* family. *Dorytomus imbecillus* was the subdominant species in the *B. syzigachne* community in 2021. *D. imbecillus* consumed plants from the genera *Salix* and *Populus* in the *Salicaceae* family [60,83]. Thus, the *Salix koreensis* community surrounding the *B. syzigachne* community can affect the abundance of *D. imbecillus*.

In 2020 and 2021, the dominant and subdominant species in the *A. indica* community were *Austroasca vittata* and *Europiella artemisiae*. Our results are consistent with previous research on insects in an *A. princeps* community in Daejeon, where *A. vittata* was the dominant species [84]. The host plant of both species is the genus *Artemisia*, including *A. indica* [42,59,85].

A match rate of over 66.7% was identified when the plants from the survey sites were matched with the known herbivorous insect host plants. This match rate is smaller than that of previous studies conducted on the Gap River, in which all three sites had a match rate greater than 90% [3]. Unlike the Gap River study, which investigated species from the riverbank to the open water area, the Miho River study examined 50 m transects within specific plant communities. Therefore, the different experimental designs may have affected the match rates of the two studies.

The sweep netting method mainly collects terrestrial herbivore insects on plants. Many Hemipteran species that use terrestrial plants as their main habitat and food source are sampled using this method. Coleopteran species, the most diverse taxa among the insect orders, were collected at a high rate in many studies [86,87,88,89,90]. Our results show that Hemipteran and Coleopteran species were higher in diversity and abundance than other orders in our target communities.

In this study, the similarities between terrestrial insect communities in the same plant community were relatively high, even for different years. The Bray–Curtis abundance similarities were significantly higher than the Sørensen similarities when the same plant community was compared between years. The insect species that were commonly abundant in the same plant community over the different years affected the Bray–Curtis similarities. The Bray–Curtis similarity of the *C. album* and *B. syzigachne* communities was high in 2020. It is likely that the index was affected by the presence of common insects, including *P. attenuata* which was the dominant species in both the communities. *P. attenuata* feeds on *H. japonicus*, which was densely distributed around both sites.

The estimated insect diversity curves with abundance revealed that when comparing the same number of individuals, the species diversity was higher in the *B. syzigachne* community than in the *A. indica* community. However, when comparing the same number of sampling units, the *A. indica* community had the highest species diversity. The sample coverage estimation based on the sampling units revealed that approximately 80% of the estimated number of species could be sampled with 50 sweeps in the *A. indica* community and 100 sweeps in the *B. syzigachne* community. Therefore, the number of sweeps required to obtain an 80% sample coverage was higher in the *B. syzigachne* community, which might explain why the Sørensen similarity of 50 sweeps (split into halves) in the *B. syzigachne* community (0.62) was lower than that of the *A. indica* community (0.8) in 2020.

The coefficient of variation for sweep variability in insect community sampling in the *B. syzigachne* and *A. indica* communities revealed that the two plant communities had similar trends in the insect orders, except for Hemiptera. Hemipteran individuals in the *A. indica* community were relatively evenly dispersed and likely affected by the high density of individuals, which resulted in the lowest coefficient of variation. However, in both plant communities, the coefficient of variation for Hymenoptera was the largest, perhaps because many Hymenopteran species (such as ants and honeybees) forage as a colony, forming a non-uniform distribution.

We investigated whether alerting by the nearby fleeing insects during sweep netting affected the sampling results by comparing two subsampling units of 50 sweeps in 2020 and 10 subsampling units of 10 sweeps in 2021 in each plant community. However, the sampling order and the number of species and individuals did not correlate. Therefore, there was no effect from insects alerting other insects during the sweep sampling in this study (Appendix A).

The sweep method for sampling insects can vary with the survey time and the weather, as well as the speed and direction of the sweeps [91]. We also found that even at the same survey site, insect communities can vary depending on the vegetation being swept.

In this study, quantitative sweep sampling in a plant community dominated by one specific plant species minimized variability with vegetation, providing basic data that could be used to establish sampling standards for each of the dominant plant communities.

The insect sampling results could not be generalized for *B. syzigachne* and *A. indica* communities in other regions, due to the limited survey sites and the environmental influence of the region. More accurate statistical results could be obtained for quantitative sweep sampling if an increased number of sites with similar climatic and environmental conditions were surveyed. 

Sweep sampling has been used in various units, from 25 sweeps to evaluate the number of agricultural pests to 800 sweeps for the diversity evaluation of tropical areas with high biodiversity [25,26]. Among the various possible numbers of sweeps that can be carried out for sampling, 100 sweeps form a common sampling unit [24,27,29,30]. Since the number of units of sweeping may vary with the target taxa range, the study area, the purpose, and the efficiency, studies standardizing and optimizing the number of units are limited [92]. Although the process of dividing the 100 sampling units into 50- or 10-sweep subsamples, along with the identification and counting of insect samples, is labor-intensive, it can evaluate the number of sampling sweeps required for an efficient environmental assessment.

Riparian zones are important habitats for potential pests and natural enemies that affect agriculture [93]. All three surveyed sites were near rice fields; hence, they can become habitats for potential agricultural pests. Recorded rice pests, such as the rice water weevil (*Lissorhoptrus oryzophilus*) and the rice stem maggot (*Chlorops oryzae*), were sampled in this study [14,94,95]. Although conservation of insect diversity and control of pest occurrence can be conflicting, most pests tend to occur in large numbers when species diversity and community stability are low and they can be more dispersed around the agricultural fields [96]. Therefore, ecological assessment and conservation around crop fields are essential for the monitoring and suppression of potential agricultural pests.

An optimized sampling method can help elucidate the relationship between the occurrence of pests in arable land and the insect community in the surrounding area. Matching insect host plants with those in the survey area may provide ecological information on the relationship between the vegetation near the agricultural fields and the insect community, including potential pests, for a more comprehensive understanding of the ecosystem and improved integrated pest management (IPM).

Riparian zones provide a mixture of artificial and natural topography with native and naturalized plants. In this environment, some plant species are translocated by people for water purification and ornamentation. Riparian zones also experience other anthropogenic interference, including farming and leisure activities, and natural interference such as sporadic flooding, which can change the vegetation and ecology of such zones [97]. Since insects have various ecological niches with short life cycles, surveying the same plant community at the same site can result in different insect communities with different sampling methods and survey periods. Insect communities can also vary due to factors such as climate, weather, species composition, season, and density of vegetation (especially annual herbs, which can be dynamic in a year). If the number of sampling sites were increased and analyzed in combination with the factors in the plant community, such as density and the defensive chemicals of neighboring plants which can affect insect communities, the relationship between the plant and insect community could then be identified more comprehensively [98].

In this study, we analyzed the insect communities according to plant communities by conducting sweep sampling with a different number of sampling units. The similarities of insect communities in the same plant community were relatively high, including similarities in different years. In addition, we obtained information to help optimize insect sampling by analyzing the sample coverage by the number of sampling units. Our research provides data on the relationship between plant and insect communities and a quantitative insect sweep sampling method for use in riparian zones. The findings of this study can assist in managing potential pests and maintaining the ecology of rivers in the future.

## Figures and Tables

**Figure 1 insects-13-00497-f001:**
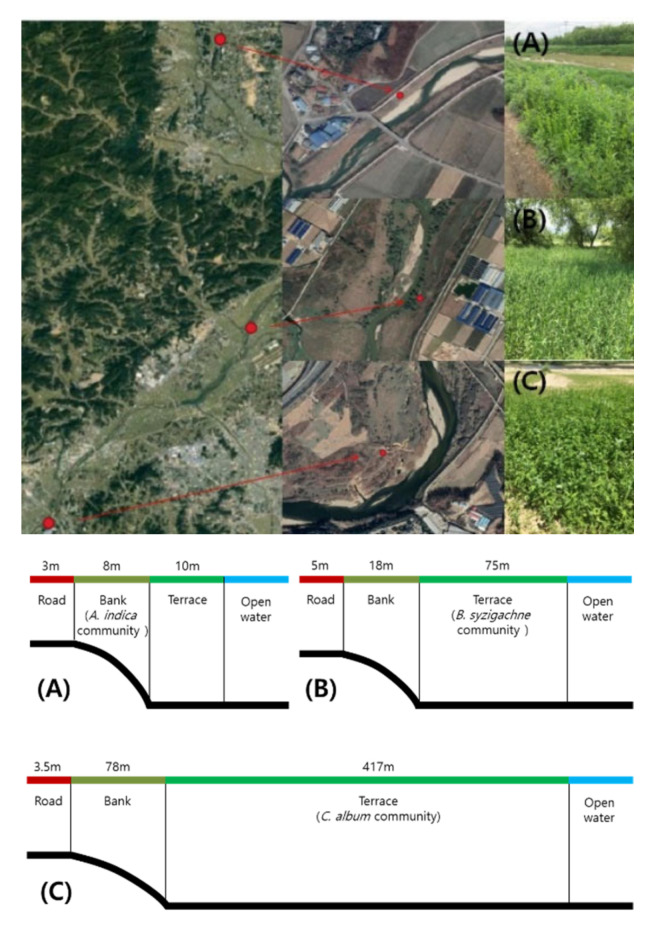
Photographs and diagrams of the survey sites and the survey methodology used in the Miho River region, Korea. (**A**) *Artemisia indica* community; (**B**) *Beckmannia syzigachne* community; (**C**) *Chenopodium album* community.

**Figure 2 insects-13-00497-f002:**
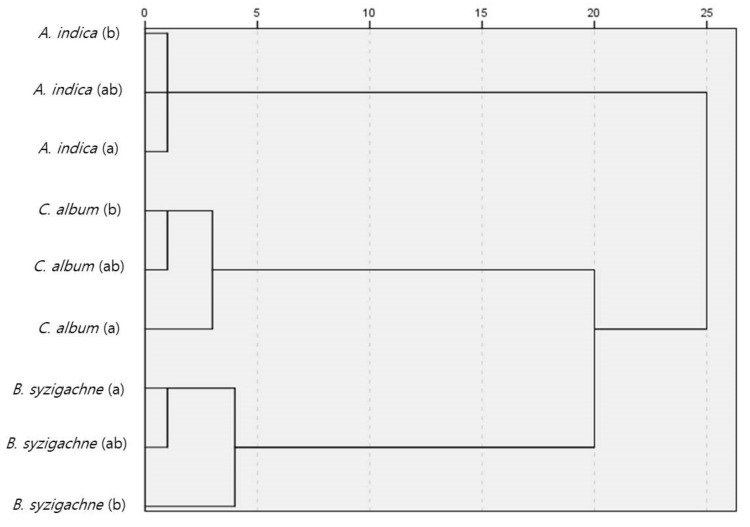
Comparison of the Sørensen similarity indices for the terrestrial insects in each plant community with sampling units from 2020. (**a**) initial 50 sweeps over 25 m; (**b**) final 50 sweeps over 25 m; (**ab**) 100 sweep total over 50 m.

**Figure 3 insects-13-00497-f003:**
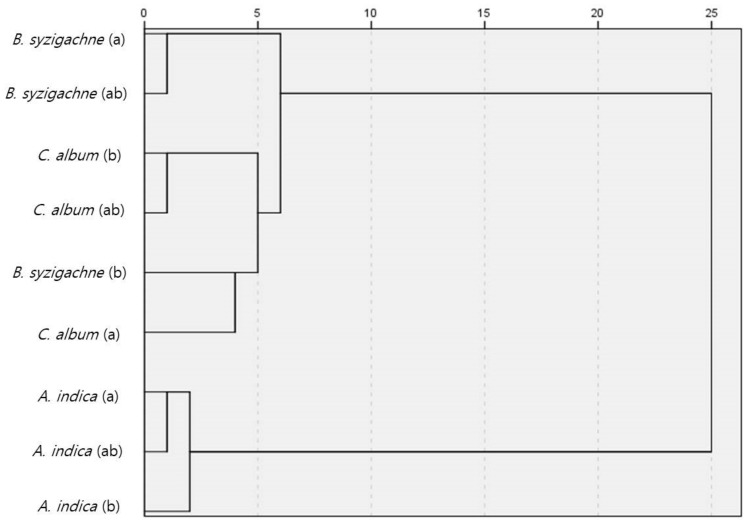
Comparison of the Bray–Curtis similarity indices for the terrestrial insects among the plant communities with sampling units from 2020. (**a**) initial 50 sweeps over 25 m; (**b**) final 50 sweeps over 25 m; (**ab**) total of 100 sweeps over 50 m.

**Figure 4 insects-13-00497-f004:**
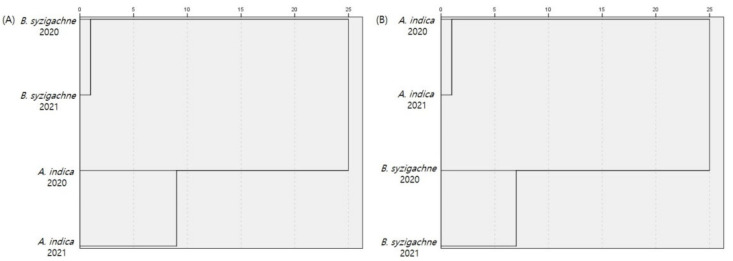
Similarity indices for the terrestrial insects among the plant communities and years. (**A**) Sørensen similarity; (**B**) Bray–Curtis similarity.

**Figure 5 insects-13-00497-f005:**
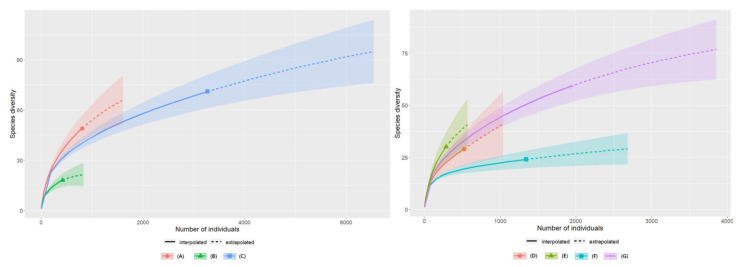
Insect species richness by plant community in 2020 and 2021. (**A**) *B. syzigachne* community in 2020, 2021; (**B**) *Chenopodium album* community in 2020; (**C**) *Artemisia indica* community in 2020, 2021; (**D**) *Beckmannia syzigachne* community in 2020; (**E**) *Beckmannia syzigachne* community in 2021; (**F**) *Artemisia indica* community in 2020; (**G**) *Artemisia indica* community in 2021.

**Figure 6 insects-13-00497-f006:**
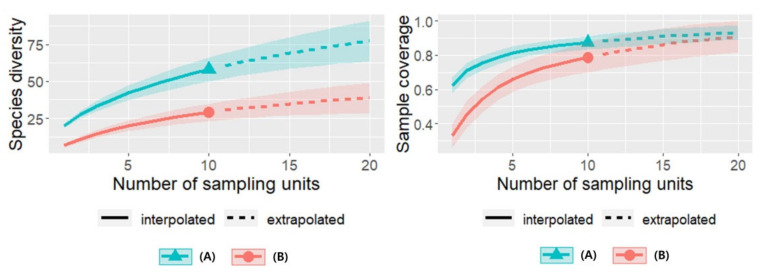
Species diversity and sample coverage of the insect sampling with the number of sampling units. (**A**) *Beckmannia syzigachne* community; (**B**) *Artemisia indica* community.

**Table 1 insects-13-00497-t001:** Results of terrestrial insect and plant surveys.

	*Chenopodium album* 2020	*Beckmannia syzigachne* 2020	*Beckmannia syzigachne* 2021	*Artemisia indica* 2020	*Artemisia indica* 2021
Individuals and species diversity of insects	2 orders 10 families 18 species 417 individuals	4 orders 15 families 29 species 520 individuals	4 orders 18 families 30 species 284 individuals	4 orders 14 families 24 species 1343 individuals	4 orders 29 families 59 species 1927 individuals
Insect *H’/DI/RI/EI	1.59/0.68/2.82/0.55	1.8/0.7/4.48/0.54	1.57/0.76/5.13/0.46	1.63/0.73/3.19/0.51	2.25/0.54/7.67/0.55
Dominant insect species	*P. attenuata*	*P. attenuata*	*C. cinctus*	*A. vittata*	*A. vittata*
Subdominant insect species	*Orthotylus flavosparsus*	*C. cinctus*	*Dorytomus imbecillus*	*Europiella artemisiae*	*Europiella artemisiae*
Diversity of plant taxa	12 families 19 genera 25 species 1 var.)	14 families 18 genera 16 species 5 var.)	10 families 13 genera 15 species 3 var.)	15 families 21 genera 20 species 2 var.)	13 families 18 genera 17 species 3 var.)
Matching rate of insect host plants with plant flora	100	82.6	66.7	87.5	66.7
Sampling completeness based on individuals	0.986	0.975	0.951	0.996	0.987

*H’, diversity index; DI, dominance index; RI, richness index; EI, evenness index.

**Table 2 insects-13-00497-t002:** Number of terrestrial insect species and individuals by order.

		*C. album* 2020	*B. syzigachne* 2020	*B. syzigachne* 2021	*A. indica* 2020	*A. indica* 2021
Hemiptera	Species	7	12	10	13	26
	Individuals	128	167	204	1160	1358
Coleoptera	Species	11	14	14	8	21
	Individuals	289	340	62	85	117
Diptera	Species	-	2	4	1	5
	Individuals	-	11	14	26	267
Hymenoptera	Species	-	-	2	2	7
	Individuals	-	-	4	72	185
Odonata	Species	-	1	-	-	-
	Individuals	-	2	-	-	-

**Table 3 insects-13-00497-t003:** Coefficient of variation of the insect communities by order in the plant communities in 2021.

	*B. syzigachne*	*A. indica*
Hemiptera	46.1	29.18
Coleoptera	43.16	43.08
Diptera	57.14	53.21
Hymenoptera	165.83	150.74
Total	36.97	35.34

## Data Availability

The data presented in this study are available in Appendix A here.

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
