# Peer review of "Sweep Sampling Comparison of Terrestrial Insect Communities Associated with Herbaceous Stratum in the Riparian Zone of the Miho River, Korea"

_insects, 2022, doi:10.3390/insects13060497_

Round 1
Reviewer 1 Report
Dear authors,
I found your manuscript very interesting. It is clearly focused and simple, and provides the necessary basic data for the conservation and management of ecologically important riparian zones in Korea through the estimation of the optimum sampling size, including the sampling rate for the estimated number of insect species. The most important aspect of your work is that it provides important data for riparian biodiversity conservation and future pest monitoring, and will certainly be useful to other entomologists in the future, especially those who conduct their research in Korea. The English language and style are really good, the statistical analysis is sound and fully supports your conclusions. Therefore, I suggest the acceptance of your manuscript for publication in its present form.
Author Response
Thank you for your comments. We are encouraged by your comments and keep working on this manuscript for publication. We appreciate you once again.
Reviewer 2 Report
You may find some suggestions as an attachment. I really enjoyed reading this pice of science, however I think it requieres some amendments to be published.

Author Response
Thank you for your comments. We have revised the manuscript as you suggested in the attached file. We appreciate you once again.
Point 1: Modification of article title (line 3)
Response 1: Thank you for the comment. We admit that ‘stratum’ is a more proper word than ‘plant communities’ in the title. As suggested, we have modified the title.
Point 2: Change of keyword (line 36~37)
Response 2: We changed keywords ‘Sweep net sampling; terrestrial riparian insects; plant communities.’ to ‘insect sampling; riparian zones; herbaceous stratum’.
Point 3: Modification of first introduction paragraph (line 40~54)
Response 3: Thank you for the comment. As suggested, we have modified the first paragraph of the introduction and cited the articles.
line 40: Riparian zones are the area of transition and interaction between the terrestrial and aquatic environments [1]. This area in its natural state is considered the most dynamic and diverse habitat in a terrestrial environment [2]. Studies have been conducted to evaluate biodiversity with insect herbivores community and specific taxa that have a high ecological role such as leaf-litter ants in the riparian zones [3–5]. Ecologically important riparian zones are mainly conserved and managed with a policy of designing and maintaining a buffer zone [6].
- Granados-Sánchez, D.; Hernández-García, M.Á.; López-Ríos, G.F. Ecología de las zonas ribereñas, Universidad Autonoma Chapingo: Texcoco, Mexico, 2006; pp. 55–69, ISSN 2007-3828.
- Naiman, R.J.; Decamps, H.; Pollock, M. The role of riparian corridors in maintaining regional biodiversity. Ecol. appl. 1993, 3, 209–212. doi: 10.2307/1941822.
- Hwang, J.H.; Kim, S.Y.; Kim, E.; Yoon, J.H.; Yim, M.Y.; An, S.L. Terrestrial insect herbivore communities of riparian zone in urban ecological river, Korea. J. Asia Pac. Biodivers. 2021, 14, 313–320. doi: 10.1016/j.japb.2021.03.009.
- García-Martínez, M.Á.; Valenzuela-González, J.E.; Escobar-Sarria, F.; López-Barrera, F.; Castaño-Meneses, G. The surrounding landscape influences the diversity of leaf-litter ants in riparian cloud forest remnants. PloS one 2017, 12, e0172464. doi: 10.1371/journal.pone.0172464.
- García-Martínez, M.Á.; Escobar-Sarria, F.; López-Barrera, F.; Castaño-Meneses, G.; Valenzuela-González, J.E. Value of riparian vegetation remnants for leaf-litter ants (Hymenoptera: Formicidae) in a human-dominated landscape in Central Veracruz, Mexico. Environ. entomol. 2015, 44, 1488–1497. doi: 10.1093/ee/nvv141.
- Graziano, M.P.; Deguire, A.K.; Surasinghe, T.D. Riparian Buffers as a Critical Landscape Feature: Insights for Riverscape Conservation and Policy Renovations. Diversity 2022, 14, 172. doi: 10.3390/d14030172.
Point 4: Modification of last introduction sentence (line 81~86)
Response 4: Thank you for the comment. We have replaced the last sentence of the introduction with a sentence representing our hypothesis in this paper.
line 83: We assumed that similarities of insect communities in the same plant community would be high even in different years.
Point 5: Joining of paragraphs in materials and methods 2.3 (line 159~166)
Response 5: As suggested, we have joined the paragraphs.
Point 6: Division of the sentence in results 3.1 (line 191)
Response 6: As suggested, we have divided the sentence.
Point 7: Addition of sampling completeness as an estimator of the inventory reliability in table 1 (line 206)
Response 7: As suggested, we have added the sampling completeness based on the individuals of each sampling.
Point 8: Change of the table 3~6 to the cluster analysis figures (line 216~253)
Response 8: As suggested, we have changed the tables to the cluster analysis figures. The tables have moved to supplementary files.
Point 9: Modification of the last sentence in results 3.4 (line 259~260)
Response 9: Thank you for the comment. As you mentioned, no significant difference is observed since 98% CI are overlapped. We have changed the sentence.
